# Emerging Methods of Monitoring Volatile Organic Compounds for Detection of Plant Pests and Disease

**DOI:** 10.3390/bios12040239

**Published:** 2022-04-13

**Authors:** Samantha MacDougall, Fatih Bayansal, Ali Ahmadi

**Affiliations:** 1Faculty of Sustainable Design Engineering, University of Prince Edward Island, Charlottetown, PE C1A 4P3, Canada; smmacdougal0@upei.ca; 2Department of Metallurgy and Materials Engineering, Iskenderun Technical University, Hatay TR-31200, Turkey; fbayansal@gmail.com; 3Department of Biomedical Science, Atlantic Veterinary College, University of Prince Edward Island, Charlottetown, PE C1A 4P3, Canada

**Keywords:** pest detection, electronic nose, volatile organic compounds, gas sensing, biosecurity, plant disease

## Abstract

Each year, unwanted plant pests and diseases, such as Hendel or potato soft rot, cause damage to crops and ecosystems all over the world. To continue to feed the growing population and protect the global ecosystems, the surveillance and management of the spread of these pests and diseases are crucial. Traditional methods of detection are often expensive, bulky and require expertise and training. Therefore, inexpensive, portable, and user-friendly methods are required. These include the use of different gas-sensing technologies to exploit volatile organic compounds released by plants under stress. These methods often meet these requirements, although they come with their own set of advantages and disadvantages, including the sheer number of variables that affect the profile of volatile organic compounds released, such as sensitivity to environmental factors and availability of soil nutrients or water, and sensor drift. Furthermore, most of these methods lack research on their use under field conditions. More research is needed to overcome these disadvantages and further understand the feasibility of the use of these methods under field conditions. This paper focuses on applications of different gas-sensing technologies from over the past decade to detect plant pests and diseases more efficiently.

## 1. Introduction

Insects and plant diseases are an essential part of the global ecosystem. Insects and diseases are often categorized into three categories: native, alien, and invasive. Native insects and diseases play an essential role in the health and regeneration of national forests. Insects and diseases introduced to new ecosystems in recent times, intentionally or not, are considered “alien”. Those who are native but spread further than their usual territory are considered “invasive” [1]. Invasive and alien species can threaten a country’s agriculture, economy, forests, and trade, as well as the livelihoods of farmers and producers across the country. Alien and invasive plant pests and diseases are introduced by the importation of plants and plant materials or the migration of different insects and other animals. Therefore, it is crucial to effectively manage plants and plant products introduced to each country to maintain and protect the global environment and economy [2].

There are currently many methods utilized across the world to detect and manage plant pests and diseases. Poland and Rassati [3] break down biosecurity surveillance into three types of activities: border surveillance, post-border surveillance and containment. Border surveillance is the act of preventing the entrance of non-native species into the importing country. Post-border surveillance and containment involve managing the spread of non-native species already within the country. Post-border surveillance is the consistent monitoring of the country’s environment for alien or invasive species and containment is the process of controlling the species once discovered. Each type of surveillance can be further broken down into specific and generic surveillance. Generic surveillance involves controlling a range of species, and specific surveillance consists of the management of a particular species. Standard generic methods of biosecurity surveillance include remote sensing [4,5], aerial surveillance [6], acoustic detection [7], genetic tools and visual inspections [8,9,10,11]. Specific methods include those listed for generic methods, as well as the addition of trained sniffer dogs and electronic noses (e-noses) [3]. Fang et al. [12] reported current, traditional, laboratory-based methods for the detection of plant diseases caused by microorganisms. These include polymerase chain reaction, fluorescence in-situ hybridization, flow cytometry, immunofluorescence, enzyme-linked immunosorbent assay and gas chromatography-mass spectrometry (GC-MS). Unfortunately, many of these methods require more time and money than is available; therefore, faster, portable, and less expensive strategies are crucial. The emerging methods, discussed ahead, all meet these requirements.

Volatile organic compounds (VOCs), such as acetone, ethanol, and tridecane, are continuously released from plant tissues and play a significant role in plant communication, competition, and defense. If a plant is damaged, it releases a profile of VOCs correlated to the type of attacker, whether the plant is healthy (has enough sunlight, water, nutrients) or has been artificially damaged, and depending on environmental conditions (temperature, humidity) (Figure 1A) [13]. VOCs allow plants to interact with insects and other plants by attracting pollinators, enemies of an attacker, providing camouflage to other plants, detecting invasive plants, signaling danger to neighbouring plants, and allelopathy (emitting an array of VOCs to affect the growth of neighbouring plants) (Figure 1B) [14]. For example, if a plant is attacked by a pest (i.e., spider mites), it releases a profile of VOCs unique to the attacker (represented by blue, red, and green molecules in Figure 1B). Nearby or neighbouring plants may then “smell” this profile, then release a profile of their own (represented by orange molecules in Figure 1B) to deter attackers (profile released in this case may resemble that of the pheromones released by enemies of the attacker) or attract enemies of the attacker (the profile released in this case may resemble that of the pheromones released by the attacker to attract enemies) [13,14,15]. The number of variables that can affect the profile of volatiles released poses a unique challenge for plant pest and disease discrimination in the field. This problem and potential solutions are discussed in Section 3.

A major step in plant volatile analysis is headspace collection. Since VOCs emitted by plants are affected by any damage inflicted on the plant, non-invasive sampling methods are required to minimize the production interfering VOCs. Tholl et al. [16,17] wrote detailed reviews on plant volatile sampling methods and separated them into two categories: static headspace sampling and dynamic headspace sampling. Static headspace sampling methods entail the use of a container or bag to enclose the plant (Figure 1C). The term ‘static’ refers to the air being static in the enclosure. Problems typically associated with static headspace sampling methods include changes in temperature and humidity in the air surrounding the plant. These changes can affect the VOCs released by the plant and may cause negative signal interferences as mentioned previously. Due to these problems, dynamic headspace sampling methods are most common. In these methods, a continuous stream of a carrier gas is moved through the headspace container, allowing for the control of temperature and humidity.

Traditionally, plant VOC “fingerprints” are monitored using GC-MS; however, GC-MS requires the use of heavy, bulky laboratory equipment that is not suitable for use in the field and is, therefore, more commonly used alongside the emerging methods of detection discussed ahead. Most applications today involve the use of electronic noses to detect pests and diseases in plants by way of VOC monitoring. An electronic nose is an array of different gas sensors (Figure 1D), combined with pattern recognition and feature extraction methods to characterize and discriminate between odours. The option exists for researchers to purchase commercial electronic noses or to develop their own devices. The latter is a more cost-effective option and allows for more flexibility in terms of use as one can chose individual sensors to suit their target gases; however, commercial options may be more robust and reliable. Measurement of sensor array response provides the user with individual responses per sensor, each with distinctive features. Figure 1E provides an example signal curve for sensors 1–8 (shown in Figure 1D). The curves provided are only an example provided to help visualize what a signal might look like, as well as feature extraction and, therefore, do not represent exactly how all sensors might react in response to being exposed to a profile of VOCs. In this case, feature 1 (F_1_) would be the slope from the exposure time (t_e_) to the final time (t_f_) and feature 2 (F_2_) would be the signal value at t_f_. Examples of other features include, but are not limited to, the slope from t_e_ to the peak value of the curve, the slope from the peak value to t_f_, or the peak value itself. The features chosen depend on the data used. Data may be normalized prior to analysis or other data analysis methods may be used to further discriminate between plant pests and diseases [18,19,20]. In Figure 1F, a sample feature plot is shown. Pathogens 1–3 can be clearly distinguished and the values for the features associated with each pathogen can be used for the further classification of unknown samples. For example, in the image, Pathogen 1 has a larger F_1_ and F_2_ versus Pathogen 2. Therefore, if an unknown sample has features comparable to Pathogen 1, it would be characterised as Pathogen 1 [18,19,20]. Some review studies have been conducted describing the signal processing and pattern recognition methods utilised in detail [18,19,20].

Previous reviews surrounding the detection of plant pests and diseases by monitoring VOCs have been written; however, some of these papers have focused solely on the use of electronic noses to detect plant pests and diseases, animal diseases, or human diseases [19,21,22]. Others have more of a focus on headspace sampling methods or are a review of the use of VOCs in agriculture, rather than for the use of plant pest and disease detection (i.e., biogenic volatile compounds for the control of pests or enhancing plant growth) [17,23,24]. This review begins with a focus on applications in detecting plant pests and diseases using emerging gas-sensing technologies, which include the use of electronic noses that take advantage of variance in electrical, gravimetric, and optical material properties, as well as applications that use other portable gas-sensing devices, such as biosensors, and field asymmetric ion mobility spectrometry from the past decade. This is followed by a review of the challenges associated with applying the technologies discussed and potential future improvements.

## 2. Emerging VOC Sensing Methods

There are a wide range of gas-sensing technologies used to detect VOCs and, therefore, plant pests and diseases. Liu et al. [25] provided a detailed description of the working principles of each technology discussed and the statistical methods commonly applied alongside them. They classify these sensing methods into two types: electrical variation (variation in electrical resistance or current) and other sorts of variation (optical sensors, gravimetric sensors, etc.).

When determining which analytical method to use for a given application, the following parameters are valuable to consider: accuracy, precision, limit of detection (LOD), selectivity, robustness, cost, response or analysis time, and ease of use. Accuracy is the degree to which a measurement agrees with an expected result. Precision is a measure of the variability between the results of multiple trials. The LOD is the smallest amount of analyte detectable by the method. Selectivity is the method’s ability to differentiate between analytes [26]. Since most of the applications mentioned in this review rely on the analysis of a profile of VOCs, rather than the detection and classification of individual compounds, the VOCs released by the pathogen in each application are only provided if they were provided in the study in question. For the same reason, individual studies use a wide array of different sensors with varied materials to be able to detect a wide array of VOCs. Therefore, details regarding the sensor materials, such as the metal oxide of choice in individual sensors, are not discussed below. Studies are available detailing individual VOCs released during plant stress and information regarding the individual sensing materials is widely available. Furthermore, only a basic description of the operating principle of each method is provided, and applications, including sampling method, data analysis methods and performances, are discussed ahead.

### 2.1. Methods with Electrical Variation

Gas sensors with variations in electrical properties are the most common methods for detecting pests and diseases in plants and plant materials. These include methods using metal oxide semiconductor (MOS) gas sensors, conductive polymer (CP) gas sensors, and electrochemical gas sensors. The gas sensors discussed in this section are most often combined with other sensors, of either the same or different types, forming an electronic nose (i.e., MOS sensors or a combination of MOS and CP sensors) to increase the selectivity of the device (Figure 1C). Upon observation, this is standard practice in the field of plant pest and disease detection by VOC monitoring; therefore, the applications discussed in this section involve the use of electronic noses.

#### 2.1.1. MOS Sensors

The most common sensor used to detect VOCs is the MOS. MOS sensors are most used in e-noses and are often used alongside GC-MS for testing and calibration purposes. MOS-type sensors are typically formed of a sensing element with sensing material coated on one side and a heater on the other. The sensing material is heated to a few hundred degrees Celsius, causing free electrons to flow through the material [27]. When the sensing material is exposed to clean air, the material oxidizes, causing a decrease in free electrons and an increase in the material resistance [27]. When exposed to a reducing gas, the gas reacts with adsorbed oxygen, releasing free electrons and decreasing the resistance of the sensing material. The change in resistivity depends on the concentration of target gas in the air [27]. Dey [28] reported that MOS sensors show excellent sensitivity, fair selectivity, excellent response time, good stability, and low cost. However, these gas sensors can be vulnerable to sensor drift caused by changes in surrounding humidity or temperature. MOS sensors have been previously used to detect bacterial infections, insect infestations and viral diseases in plants.

Biondi et al. [29] used the PEN3 Electronic Nose (Airsense Analytics, GmBH, Schwerin, Germany) to detect potatoes infected with *Ralstonia solanacearum* or a subspecies of *Clavibacter michiganensis* (causes of potato brown rot and ring rot, respectively). Volatile markers for brown rot include short-chain alcohols and ketones, and 3-methyl-2-pentanone has been identified as a marker for ring rot. Both static dynamic sampling methods were used. Linear discriminant analysis (LCA) and principal component analysis (PCA) were used to analyze samples. The study correctly classified 81.3 and 57.4% of the samples for static and dynamic sampling methods, respectively [29]. Xu et al. [30] also used the PEN3 e-nose to detect the age and amount of brown rice plant hoppers (*Nilaparvata lugens*). The team used static sampling methods coupled with data analysis methods, such as PCA, LDA, probabilistic neural network (PNN), back-propagation neural network (BPNN) and loading analysis (Loadings). The discrimination accuracies of age and amount were 96.67% and 64.67%, respectively, when PNN was used and 96.67% and 47.33%, respectively, when BPNN was used. PEN3 was also used by Rizzolo et al. [31] to detect *Rhynchophorus ferrugineus* in palms, with classification rates of up to 100%. Static sampling methods and PCA and DA methods were used to analyze the data. Cellini et al. [32] discriminated between fire blight infected, blossom blight infected, mock-infected and control groups of apples, using two different MOS sensor arrays: the EOS507C Electronic Nose (Sacmi, Imola, Italy) and the PEN3 e-nose. Both sensors proved capable of discriminating between infected, mock-infected, and control groups; however, the EOS507C was the only one that could distinguish between the two infections. The reason provided for this is that the EOS507C contains a humidity control system that allows the user to set a dew point, where the PEN3 simply has a maximum humidity of 95% and may not function properly near the limits. Static sampling methods were combined with PCA and LDA, processed to facilitate discrimination. Rutolo et al. [33] used an array of 12 MOS sensors to detect soft rot in potatoes caused by *Pectobacterium carotovorum*. The study showed the array was capable of early detection with selectivities and sensitivities of 80–100%. The team used dynamic sampling methods and used a wide range of feature extraction and data analysis methods. VOC markers for soft rot have been determined to be acetone, ethanol, 2-butanone, and 3-hydroxy-2-butanone [29]. In 2017 and 2019, Sun et al. [34,35,36] showed that the discrimination between *Ectropis obliqua* and *Ectropis grisescens* infections of different invasive times, severities, and ratios in tea plants was possible using the PEN2 e-nose system (an MOS-based e-nose; Airsense Analytics, GmBH, Schwerin, Germany), feature extraction and LCA. Discrimination rates as high as 100%, 96.90% and 93.75% were obtained in these studies, respectively. They also successfully used the PEN2 system to discriminate between different damage types in tea plants, with discrimination rates as high as 100% using feature extraction and data analysis methods, such as PCA, locality preserving projections (LPP), and support vector machine (SVM). The highest discrimination rates occurred with the combination of LPP and SVM. The early detection of *Botrytus cinerea* in tomato plants using the PEN2 e-nose system was shown to be feasible, with a discrimination rate of 100% with the use of kernel PCA and LCA methods [37]. Mishra et al. [38] used the Fox 4000 e-nose system (Alpha MOS, Toulouse, France) an MOS-based e-nose to accurately detect *Sitophilus granarius* infestation in stored wheat grain, using fuzzy logic and statistical analysis methods, such as PCA. Wen et al. [39] developed a Sweeping Electronic Nose System (SENS, a MOS sensor array) to detect Hendel (*Bactroicera dorsalis*) infestation in citrus fruits, with recognition rates as high as 100% using PCA and LCA. In 2019, Cui et al. used a four MOS-based sensor array to detect *Myzuz persicae*-infested tomato plants. Fast sensor response and high sensitivity were reported. The team used static sampling methods and PCA data analysis [40]. Wang et al. [41] successfully used the PEN3 e-nose system to detect wood-boring insect *Semanotus bifasciatus* infestations of different durations in *Platycladus orientalis*, an evergreen tree native to eastern Asia. The team achieved classification rates as high as 99.80%. Static headspace sampling methods were used alongside PCA, combined with grid-search SVM. In 2021, a low-cost, six MOS-based electronic nose was tested and proved to be capable of accurate discrimination of *Pythium intermedium* and *Phytophthora plurivora* using feature extraction and SVM [42].

MOS sensors may also be used to detect viral and fungal infections. The PEN3 e-nose was used for the detection and classification of fungal diseases (*Botrytis* sp., *Penicillium* sp., and *Rhizopus* sp.) in post-harvest strawberries, with 96.6% accuracy [43]. In this study, Pan et al. collected samples using static headspace sampling methods and PCA for data analysis. Jia et al. [44] used the PEN3 e-nose coupled with dynamic headspace sampling methods and LCA, SVM and BPNN to differentiate between fresh apples and apples infected with *Aspergillus niger* and *Penicillium expansum,* with recognition rates as high as 90.0%. The PEN2 e-nose was also used for the detection and classification of three species of *Aspergillus* in rice kernels, with accuracies as high as 96.4% using dynamic headspace sampling methods and PCA, SVM and BPNN data analysis methods [45]. Suchorab et al. [46] used an MOS-based e-nose for the detection of fungal-infected building materials, with 80–85% accuracy. Samples were analyzed using PCA. Nouri et al. [47] reported a fast, reliable, and non-destructive technique for detecting *Alternaria* spp.-infected pomegranate and accuracy as high as 91.67%. Samples were collected using dynamic headspace sampling and analyzed using LDA, SVM and BPNN. Hazaika et al. [48] used the Alpha MOS Fox 3000 (a MOS sensor array; Alpha MOS, Toulouse, France) to discriminate between plants mildly and moderately infected with Citrus Tristeza Virus, with a discrimination accuracy as high as 97.67%. Kresnawaty et al. [49] used an 8 MOS sensor array to detect and distinguish between species of *Ganoderma* fungal infection in oil palm, with accuracies of 99.64%. Samples were collected using dynamic headspace sampling and SVM analysis methods were used. Oates et al. [50] used a low-cost e-nose, consisting of eight MOS sensors, to detect lethal bronzing disease in cabbage palms (*Salbal palmetto*). Samples were collected using dynamic headspace sampling methods. The device was able to discriminate between healthy and infected plants using PCA. 

#### 2.1.2. CP Sensors

Conductive polymer gas sensors consist of a mixture of polymers and conductive materials, such as carbon black deposited over two electrodes [51]. When the polymer encounters target gases, the polymer swells reversibly, creating distance between the conductive polymer particles (i.e., a decrease in the concentration of carbon black), thus, changing the electrical resistance of the material [52]. In 1997, the Bloodhound BH114 (conductive polymer-based device; Bloodhound Sensors, Leeds, UK) was used to detect a range of microorganisms by analyzing the volatiles produced by each microorganism, using static headspace sampling. The sensor proved capable of distinguishing between thirteen types of bacteria and three types of yeast, with classification rates of up to 100% using PCA [53]. In 2008, the VOC profiles emitted by cucumber, pepper and tomato plants were analyzed to detect damage from pests, diseases and artificial damage using the Bloodhound ST214 (13 CP array: Scensive Technologies Ltd., Normanton, UK). The e-nose was able to successfully discriminate between spider-mite-infested, undamaged, and artificially damaged cucumber leaves. It was also successful in discriminating between powdery-mildew-infected, tobacco-hornworm-infested, artificially wounded, and unwounded control groups of tomato leaves, as well as undamaged leaves of pepper, tomato, and cucumber plants. Laothawornkitkul et al. [13] reported high discrimination power, rapid response, and low detection limits for the Bloodhound ST214 e-nose. Li et al. [54] used Cyranose 320 (32 CP sensor array; Sensigent Intelligent Sensing Solutions, Baldwin Park, CA, USA) to detect sour skin in onions caused by *Burkholderia cepacian*, using PCA and SVM analysis methods. The study reports excellent selectivity, fast response times (2 min per sample) and high accuracy; however, the study also reports a false positive rate of 30%. In the same year, Li et al. [55] used the Cyranose 320 system to classify grey mould, anthracnose, and *Alternaria* rot in blueberry fruits, three post-harvest diseases caused by microorganisms. The team reported an overall accuracy of 90% using PCA and good reversibility; however, some sensor degradation over time was evident. Henderson et al. [56] used Cyranose 320 to detect stink bug presence and stink bug damage on cotton bolls. The e-nose could discriminate between the volatiles produced by the southern green stink bug and the green stink bug (*Chinavia halaris*) using principal component analysis. Although the e-nose itself returned confused responses, it could detect stink bug damage on cotton bolls with a prediction accuracy of 90%. Li et al. [57] used a 32-sensor array to detect onion sour skin and botrytis neck rot, with a 97.8% classification rate using PCA and a short sampling time of 2 min. The Cyranose 320 was also used to detect stink bugs by analyzing the VOCs emitted from cotton bolls with dynamic headspace sampling methods and discriminating between two stink bug species (southern green stink bug, *Nezara viridula*, and brown stinkbug, *Euschistus servus*) using PCA. The e-nose proved capable of differentiating between control and damaged cotton bolls, with an accuracy of 87.5%; however, the success rate for differentiating between which species the cotton bolls were attacked by was much lower, at 65%, indicating the e-nose used in this study was not capable of species-specific discrimination [58]. Ghaffari et al. [59] used a 13 CP sensor array to detect and discriminate between *Tetranychus urticae*-infested and *Oidium neolucopersici*-infected cucumber, pepper, and tomato plants, with accuracies of up to 96%. Samples were collected using static headspace sampling methods and SVM data analysis tools were used. Gruber et al. [60] constructed a 4 CP e-nose to detect *Penicillium digitarium* in oranges using dynamic headspace sampling methods. After only one day post-inoculation, the e-nose was able to discriminate between infected and healthy oranges using PCA. The authors report good portability, low power consumption, low cost, a lifetime of over one year and short response time (~5 min); however, the authors also report a potential issue with interference from insecticides and other potential fumigants. Wilson [61] reported high accuracy in the discrimination and classification of bacterial wetwood disease in American beech and black cherry trees using the Aromascan A32S (32 CP array). Samples were collected using static headspace sampling methods and data were analyzed using PCA and Quality Factor statistical values. Lampson et al. [62] constructed a CP e-nose to detect Kudzu bugs, a soybean pest. The VOC fingerprints released by the Kudzu bugs were analyzed and correctly classified, with accuracies of up to 94.4%. The group noted that only 1 s sampling time is needed to detect the pests and that further study on the feasibility of detection in field conditions is needed. 

#### 2.1.3. Electrochemical Sensors

Electrochemical sensors typically detect oxygen and toxic gases, such as carbon monoxide. The gas-sensing layer is composed of a working electrode, a counter electrode, an electrolyte, and a reference electrode. Target gases diffuse into the gas-sensing layer and undergo oxidation or reduction reactions with the electrolyte, causing current to flow either from the working electrode to the counter electrode or vice versa. The current produced is proportional to the concentration of target gas [63,64]. Rutolo et al. [65] used the WOLF 4.1 (9 electrochemical sensor array) to detect soft rot in potatoes. Samples were collected through dynamic headspace sampling methods. The sensor array combined with linear discriminant analysis showed 100% specificity and sensitivity, in both symptomatic and pre-symptomatic groups. The study also reports low cost, low power consumption, good tolerance to environmental changes, and operation at room temperature. 

### 2.2. Methods with Variance in Other Properties

Many methods for detecting pests and diseases in plants by monitoring VOCs exist that depend on variance in physical properties other than electrical properties. These include gravimetric methods (such as those which require the use of quartz crystal microbalance (QCM sensors), optical methods (such as colorimetric sensors), biosensing methods, and other methods (such as field asymmetric ion mobility spectrometry (FAIMS)). QCM sensors work by the attachment of two gold electrodes to a quartz crystal with a semi-selective coating that generates an alternating current, producing oscillation of the quartz at a fundamental frequency. Detection functions by measuring the change in frequency caused by an added mass due to adsorbed material. The frequency shift is proportional to the mass of the added material [66]. Colorimetric sensors detect VOCs using nanomaterials and chemo-responsive organic dyes, which interact with the target compounds and produce a detectable color change. An array of dyes and materials that react to different compounds and changing colour, can create a “fingerprint” specific to different odors [67,68]. Biosensors are devices that use a biological sensing element (i.e., enzymes, cells, etc.) and can detect chemical reactions by the formation of signals proportional to the target substance [69]. FAIMS functions by first ionizing the VOCs released by the plant. The ions are then sent through an electrode channel with a ranged compensation voltage and an applied RF waveform that induces the separation of the ions based on the different ion mobilities. Each ion then hits the detector at a different compensation voltage with a certain ion current. These values are plotted to provide the user with a unique profile for each ion [70,71,72].

In 2014, Rutolo et al. used FAIMS to detect soft rot in stored potatoes using dynamic headspace sampling and PCA, reporting high sensitivity and achieving accuracies of 90% [71]. In 2017, Sinha et al. also used FAIMS to detect sour skin caused by *Burkholderia cepacia* in stored onions and soft rot caused by *Pectobacterium carotovorum* in stored potatoes. The device could detect sour skin within three days post-inoculation and soft rot within two days post-inoculation, with accuracies of at least 83% using PCA and other classification models [72,73]. In 2018, Fang et al. produced a tri-enzyme electrochemical biosensor to detect methyl salicylate, a volatile biomarker for indication of plant stress. The study reports high sensitivity and a low limit of detection [74]. Li et al. [75] developed a smartphone-based VOC sensor for the diagnosis of late blight in tomato plants. This sensor system consisted of a disposable colorimetric sensor array. The study reported 100% sensitivity, 90% specificity and 95% accuracy in lab conditions and 95% sensitivity, 100% selectivity and 97.5% accuracy in field conditions. Wang et al. [76] used a QCM sensor array to detect invasive bark beetles and juniper bark borers in *Pladycladus orientalus*. The paper reported good sensitivity, selectivity, and stability over at least one month. The sensor array produced high classification rates, with only one misclassification during the study. Chalupowicz et al. [77] developed a whole-cell-based biosensor, taking advantage of luminescent changes in bacteria due to changes in VOCs, and tested the device for the detection of *Penicillium digitatum* in oranges. Results of the study showed the feasibility of using these whole-cell-based biosensors; however, more studies are needed to further determine the characteristics of the sensors themselves. Wen et al. [78] used a QCM sensor array coated with ethyl-cellulose for the detection of d-limonene, a marker for *Bactrocera dorsalis* infestation in citrus fruits. The study reported repeatable results, with a determination coefficient of 0.9899. The applications, advantages and disadvantages of each method described in Section 2 are summarized in Table 1.

## 3. Challenges and Future Improvements

Although studies have shown that discriminating between healthy, damaged, or infected plants is possible, there are many challenges one might face when applying these technologies in the real world, and improvements that may be made to combat these difficulties.

The first challenge, and likely the most debilitating, when it comes to further implicating the analysis of VOCs for plant pests and disease detection is the reactive nature of VOCs emitted from the plants and the sheer number of variables present, which can affect the VOC profiles emitted. In the lab, it is much easier to control variables, such as temperature and humidity. In the field, this becomes much more challenging; therefore, inconsistent VOC profiles between plants of the same species in the field, even healthy, pose a unique challenge for disease diagnosis. For instance, healthy plants under environmental stress may present similar profiles to diseased or damaged plants, making it difficult to discriminate between them [19,79]. Plants release VOCs in response to disease or damage due to pests or animals; however, they also release VOCs due to environmental factors, such as too much light, high or low temperatures, lack or excess of water, lack of oxygen, nutrient deficiencies, humidity, and more [79]. VOCs emitted by the plants also change throughout their life cycles [19]. Therefore, the same species grown in various locations, soils, or environments, or the same species at various stages in their life cycle, will have different VOC profiles [19,79]. Further studies are needed to continue building databases that include each species of plant, information regarding soil type, location, weather, and available nutrients [24]. Another potential solution may be to focus on the VOCs emitted by the pests or infections themselves, rather than the VOCs emitted from the plants. As mentioned previously, Henderson et al. [56] used an electronic nose to detect the presence of stink bugs by detecting the VOCs emitted by the bugs themselves. Senthilkumar et al. [80] used GC-MS to characterize the VOCs emitted by *Tribolium castaneum* and *Cryptolestes ferrugineus*. Xu et al. [30] used the PEN3 electronic nose to determine the age and amount of brown rice plant hoppers by sensing the VOCs emitted by the bugs themselves. Barbosa-Cornellio et al. [81] wrote a review discussing sampling and analysis methods for VOCs from insects. Potential future studies may include the detection of any of these pests by the detection of their emitted VOCs, using any of the methods mentioned in this review.

Challenges also exist regarding the methods themselves. Many of the methods discussed above are sensitive to humidity, temperature, or atmospheric pressure. Some attempts in mitigating this issue might include the addition of a humidity control system, such as that added to the EOS507C MOS sensor system [28]. Paknahad et al. [82] developed a humidity removal membrane for microfluidic-based gas-sensing devices. Further additions to systems suffering from sensitivity to environmental factors might include temperature regulating systems or pressure-regulating systems. Furthermore, many of the methods discussed in this review require the use of a sensor array, containing up to thirty-five different sensors, each with its own selectivity. This requires a more complicated analysis process and more frequent calibration due to individual sensor drifts. It also calls for more frequent replacements of individual components in the system. Therefore, a more reliable and straightforward version of electronic noses is required. Unfortunately, most sensors are not very selective on their own. Previously, gas sensors have been combined with microfluidic channels to allow component separation along the channel before the sample reaches the sensor. This type of e-nose only requires one sensor and acts based on the diffusion properties of the compounds involved, similar to GC-MS (see Figure 2) [83,84,85,86,87,88,89,90]. Paknahad et al. [91] suggested and tested the use of microfluidic-based e-noses, combined with feature extraction methods, to successfully discriminate between several types of wine. A similar method could potentially be used for the detection of plant pests and diseases.

Many of the devices mentioned above have been tested in the lab, but not in the field. To further understand how the devices will function in the real world and their true feasibility, the methods need to be tested for use in the field. This may involve testing the devices at the border, alongside currently used methods or on crops. One challenge with field testing is the difficulty associated with sample collection. As mentioned previously, traditional volatile sampling methods can be invasive and may cause unwanted changes in VOC profiles due to damage or stress to the plant. Sampling VOCs from the headspace of a plant is a non-invasive alternative to such methods. Static headspace sampling methods of VOC trapping include headspace enclosure, where a chamber or bag is used to passively collect VOCs emitted from the plant. The chamber or bag can impact temperature, humidity, light, and other factors affecting the VOCs emitted from the plant. Other methods include solid-phase microextraction, where VOCs can be collected from both foliage headspace and soil headspace [92,93]. Direct contact sorptive extraction, or stir-bar sorptive extraction, is used as a more adsorbant alternative to SPME, where a stir-bar coated with PDMS is held on to the surface of foliage with a magnet, to adsorb volatiles directly from the leaf or from the air around it. The PDMS then undergoes thermal desorption and is coupled with an analytical method, such as GC-MS [94]. Another challenge associated with field testing is the presence of interferences due to outside factors and background noise. Solutions include the development and use of new statistical analysis methods, such as signal processing, and data analysis tools, such as PCA, LDA and neural networking, as mentioned previously.

## 4. Conclusions

Plant pests and diseases can cause the widespread loss of crops and destruction of global ecosystems. Therefore, it is essential to be able to control and monitor the introduction and spread of such insects within each country. Traditional methods of doing so require training and expertise and are often expensive and bulky. Therefore, simpler, cost-effective methods are needed. Methods that meet these requirements have been discussed and include electrical gas-sensing technologies and other gas-sensing technologies that take advantage of volatile organic compounds released by plants under stress. These methods are less expensive to purchase and produce, faster, and easier for the public to use than traditional laboratory methods. Most commonly, dynamic headspace sampling methods are used and PCA, SVM and BPNN appear to be the most common and effective data analysis tools. Determining which detection method to use depends on the environment it will be used in, as well as other factors specific to the application (i.e., volatiles being detected, type of plant, pest, etc.). Some difficulties associated with each of the gas-sensing methods discussed include the reactive nature of VOCs and the sheer number of uncontrollable variables present in the field, sensitivity to environmental factors, and sensor drift. Furthermore, most studies presented in this review were conducted in the lab. Further studies in the field are needed to resolve these challenges and determine the feasibility of using these methods to detect pests and diseases and maintain the health and diversity of the global ecosystem.

## Figures and Tables

**Figure 1 biosensors-12-00239-f001:**
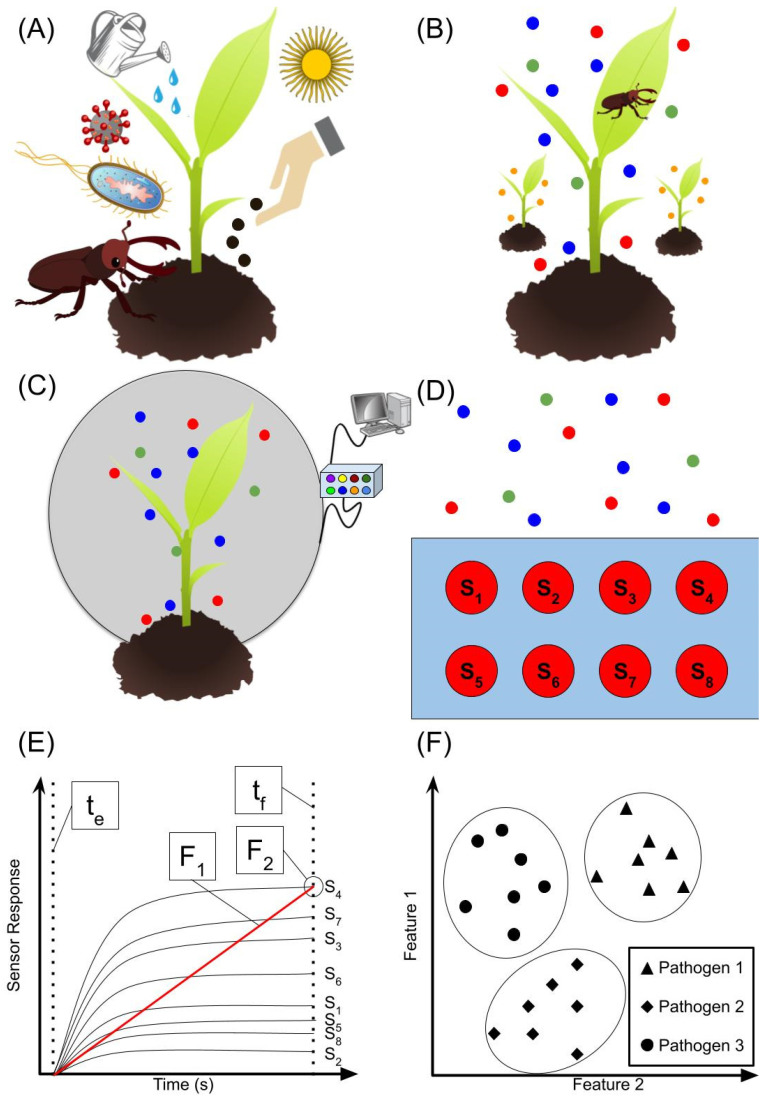
Overview of plant pest and disease detection by way of VOC monitoring using an electronic nose for example. (**A**) Effects of external factors on VOCs. VOCs such as acetone and ethanol are continuously released from plant tissue due to external factors such as temperature, available nutrition, or damage by pests or viral and bacterial infections [13]. (**B**) VOCs for plant communication and defense. VOCs released by plants allow communication with insects and other plants (i.e., attracting pollinators or signaling danger to neighbouring plants) [14,15]. (**C**) VOC collection. The profile VOCs released by plant tissues at any given moment can be collected using a headspace sampling method such as static headspace sampling (shown here) or dynamic headspace sampling [16,17]. (**D**–**F**) Basic working principle of an electronic nose. (**D**) An array of different sensors (S_1_–S_8_, each selective to different analytes) exposed to headspace sample. (**E**) Example of a response curve from an electronic nose with eight sensors (S_1_–S_8_) from exposure time, t_e_, to final time, t_f_, (i.e., 200 s) with features 1 and 2 (F_1_, F_2_). (**F**) Sample feature plot for three pathogens (1–3) that affect the same plant (such as *Ralstonia solanacearum*, *Clavibacter michiganensis*, and *Pectobacterium carotovorum*, causes of different types of rot in potatoes).

**Figure 2 biosensors-12-00239-f002:**
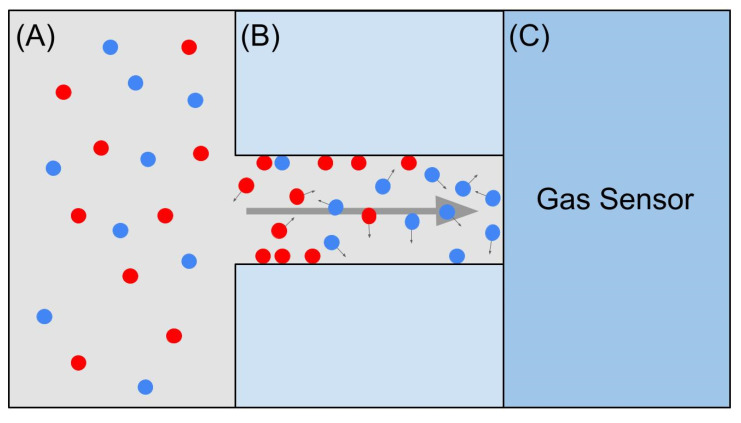
Basic operating principle for microfluidic electronic nose devices. Unlike electronic noses discussed in Section 2, the sensor array has been replaced with a single sensor coupled to a microfluidic channel. (**A**) Headspace sample containing a mixture of so analytes, red and blue. (**B**) Sample flows through microfluidic channel due to diffusion processes. Separation occurs due to adsorption on the channel walls. (**C**) Compound with the lowest retention time (time spent within the channel) will reach the gas sensor. This provides the user with a curve with features specific to the mixture of gases within the sample allowing for discrimination [84].

**Table 1 biosensors-12-00239-t001:** Summary of applications, advantages, and disadvantages of gas-sensing methods as reported in the applications discussed in this review.

Sensing Mechanism	Applications	Advantages	Disadvantages
MOS	*Pectobacterium carotovorum* [33] *Ectropis obliqua* [34,37,39] *Sitophilus granarius* [36] *Ectropis grisescens* [37,38] *Bactrocera dorsalis* [39] *Myzus persicae* [40] *Semanotus bifasciatus* [41] *Phytophthora plurivora* [42] *Pythium intermedium* [42] *Botrytus* sp. [43] *Penicillium* sp. [43,46] *Rhizopus* sp. [43] *Penicillium expansum* [44] *Aspergillus* sp. [44,45,46] *Paecilomyces* sp. [46] *Acremonium* sp. [46] *Ganoderma* spp. [47] Citrus Tristeza Virus [48] *Altenaria* spp. [49] Lethal Bronzing Disease [50]	high sensitivity [19] cross-sensitivity [35] fast [19,49] reliable [49] low cost [49] non-destructive [49]	sensitive to sulphurs [10] high temperature [19] low selectivity [26] sensitive to environmental factors [26]
CP	*Manduca sexta* [13] *Escherichia coli* [53] *Pseudomonas aeruginosa* [53] *Citrobacter freundii* [53] *Enterobacter aerogenes* [53] *Bacillus cereus* [53] *Klebsiella aerogenes* [53] *Candida albicans* [53] *Staphylococcus aureus* [53] *Staphylococcus epidermis* [53] *Salmonella reading* [53] *Salmonella poona* [53] *Salmonella garinarium* [53] *Bacillus subtillis* [53] *Burkholderia cepacia* [53,57] *Botrytis cinerea* [55] *Colletotrichum gloeosporioides* [55] *Altenaria* sp. [55] *Chinavia halaris* [56] *Nezara viridula* [56,58] *Botrytis allii* [57] *Euschistus servus* [58] *Tetranychus utricae* [13,59] *Oidium neolycopersici* [13,59] *Penicillium digitatum* [60] Bacterial Wetwood [61] *Megacopta cribraria* [62]	high sensitivity [25] low cost [25] short response time [19,25] low energy consumption [25] portable [25]	unstable [25] poor selectivity [25] sensitive to environmental factors [25]
Electrochemical	*Pectobacterium carotovorum* [65]	high selectivity [64,65] fast response time [64,65] room temperature operation [65] low power consumption [64,65] low cost [64,65] low limit of detection [65] tolerant to environmental factors [65]	limited operational lifetime [64] poor baseline stability [64]
Colorimetric	*Phytophthora infestans* [69]	disposable [19] portable [19,75] fast response [19] robust [19]	sensitive to humidity [19]
FAIMS	*Pectobacterium carotovorum* [71,73] *Burkholderia cepacia* [72]	operates at atmospheric pressure [70] inexpensive [70] high sensitivity [71] tolerance to environmental conditions [71] portable [71,72,73]	lower performance [70] lower accuracy [70]
Biosensing	*Methyl salicylate* [74] *Penicillium digitatum* [77]	real-time [19] high specificity [19]	unstable [19] sensitive to pH [19] sensitive to environmental factors [19]
QCM	*Semanotus bifasciatus* [76] *Phloeosinus aubei Perris* [76] *Bactrocera dorsalis* [78]	work at room temperature with high sensitivity [19] Long lifetime [25]	poor reproducibility [19] low sensitivity [25] sensitive to environmental factors [25]

## Data Availability

Not applicable.

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
