# Peer review of "Emerging Methods of Monitoring Volatile Organic Compounds for Detection of Plant Pests and Disease"

_biosensors, 2022, doi:10.3390/bios12040239_

Round 1
Reviewer 1 Report
The manuscript is interesting, it makes an exhaustive review of sensors to measure VOCs.
I would make a review of the English language, a comparative cost analysis between new sensors and traditional methods could be added and it would be necessary to deepen in the conclusions.
Reviewer 2 Report
The manuscript by S. MaDougall et al. titled “ Emerging methods for monitoring volatile organic compounds for detection of plant pests and disease: a review” presents a review of the electronic noses applications to the detection of plant infestation or disease. Though this is an interesting topic, this review has several deficiencies.
Rationale for writing that review is not clearly explained. On lines 71-78 the authors state that previous reviews “focused solely on the use of sensor arrays or…”, but the present review “focuses solely on the emerging methods …. to detect plant pests and diseases”. In fact this manuscript describes solely e-noses with exception of one work that used biosensor, therefore it is not clear what differentiates this review from the ones published recently, e.g. ref. 16-18.
Discussion of the gas sensors is presented in very generic terms and is not very informative. As numerous reviews, articles and textbooks exist that describe operating principle and materials employed in the gas sensors, references would be sufficient.
Another issue is that the applications of the e-noses to the disease and pest detection are listed without any critical discussion, which limits this review interest and usefulness for the readers. No information about the main VOCs emitted by the infected or infested plants is presented. No information about sensing materials used for each type of sensors is presented. Consequently, no information nor discussion of the appropriate sensor types and sensing materials for each type of analysis is presented. For instance, MOS and CP based gas sensors have different sensitivity and selectivity, and both types were used for detection of infestation with insects and microorganisms, but it is not clear why particular type of sensors was selected for the particular task.
Lines 87-97 and Fig. 3. Please use conventional nomenclature, i.e. gas sensing methods => transducers or measuring principles in this context; variation or variance is not used to describe measuring principles, i.e. should be electrochemical sensors, optical sensors and mass sensors.
English language needs extensive editing to facilitate reading.
Reviewer 3 Report
Review:
Ref.: biosensors-1589824
Title: Emerging Methods for Monitoring Volatile Organic Compounds for Detection of Plant Pests and Disease: A Review
Authors: Samantha MacDougall , Fatih Bayansal , Ali Ahmadi *
This review addresses the uses of sensor for the detection of plant pests and disease. The authors consider only sensors that induce an electrical variation such as Metal-Oxide-Sensors (MOS), Conductive Polymer sensors (CP) and finally Electrochemical Sensors.
The authors explain how these sensors are formed and but they do not explain the principles of such sensors. For clarity purpose, schematics should be added for each type of sensors followed with a short explanation of the principle. In addition, differences between commercial sensors and some sensors developed by academic researchers should be made. Moreover, the statistical approach used for the data analysis is not provided.
Tables should be better organized. It would be more interesting to have a column with the applications, the type of sensor material used if available (what metal oxide for example); and in place of advantages and disadvantages, sensitivity, detection limits and limits of use, in the observed application should be reported.
The conditions of volatiles trapping should be mentioned for the applications discussed in this paper.
When the same author is, cited twice for the same application, with the same sensor (Sun et al, line 132, 143) more details regarding each publications should be added or the references should be combined
Reviewer 4 Report
Dear Authors;
I think you have presented a good review paper deals with type of sensors, method of detection, and gives principles of sensors works as well as advantages and disadvantages of each type; but I have some note and I ask you to make the required changes, which are:
1- section 2.1.1 MOS Sensors, I think you have to list the literature review papers ascending order
(Date of publication).
2- For Section 2.1.2 CP Sensors, you have to write the previous works as follows:-
Li 2009, Li 2009, Henderson 2010, Li 2011, Gruber 2013, Wilson 2014, and Lampson 2017.
3- for the Section 2.2 Methods with variance in other properties, you have to reorder the papers as follows:-
Fang 2018, Li 2019, Wang 2020, Chalupowicz 2020, and Wen 2021
Round 2
Reviewer 2 Report
Most of the issues raised in the previous review have been addressed by the authors and the quality of the manuscript has improved.
All methods described in this review can be classified as e-noses. Thus, description of the scope of this review and in particular difference between this and recently published reviews on the application of the e-noses to plant disease detection needs to be further clarified.
